# GRADIENT STORM: STRONGER BACKDOOR ATTACKS THROUGH EXPANDED PARAMETER SPACE COVERAGE

## ABSTRACT

Targeted data poisoning poses a critical adversarial threat to machine learning systems by enabling attackers to manipulate training data to induce specific, harmful misclassifications. Among these threats, backdoor attacks are particularly pernicious, embedding hidden triggers in the data that lead models to misclassify only those inputs containing the trigger, while maintaining high accuracy on benign samples. In this paper, we propose Gradient Storm, a novel technique that facilitates the simultaneous execution of multiple backdoor attacks, while necessitating only minimal modification to the training dataset. Our contributions are twofold: First, we introduce a method for designing adversarial poisons in modular components, each tailored based on a distinct region of the model's parameter space. Second, we present a framework for conducting multi-trigger attacks, where each trigger causes misclassification from a specific source class to a distinct target class. We evaluate the efficacy of Gradient Storm across multiple convolutional neural network architectures and two benchmark datasets, demonstrating its robustness against eight different poisoning defense mechanisms. Additionally, we show that poisons crafted for one model can be effectively transferred to other models, demonstrating that our attack remains effective even in black-box settings.

## 1 INTRODUCTION

Deep Neural Networks (DNNs) have been successfully employed in various fields, including computer vision, speech recognition, and natural language processing (e.g., Redmon & Farhadi (2018); Baevski et al. (2020); Brown et al. (2020)). Despite their high performance, training or fine-tuning these networks often requires task-specific datasets and significant processing power, which may not always be accessible to all users. As a consequence, users may rely on third-party providers for obtaining the necessary resources. This dependency introduces a risk where adversaries could embed malicious samples into the datasets, leading to models that exhibit unintended behaviors, such as misclassification. These attacks are known as *data poisoning*, as described by Muñoz-González et al. (2017). Backdoor attacks, introduced by Gu et al. (2019), represent a specific form of data poisoning designed to embed a trigger within the model during the training phase. The resulting model behaves normally on benign inputs but misclassifies any image containing the trigger as the target class during inference. A notable demonstration by Liu et al. (2018b) in a driving simulator environment showed how a roadside billboard used as a trigger could cause an autonomous driving system to fail, resulting in accidents.

Early methods to incorporate triggers into models involved patching source class images with triggers and re-labeling them with a target class label (Gu et al. (2019); Chen et al. (2017)), making them easily identifiable by human observers. To visually hide triggers, Barni et al. (2019) added an invisible sinusoidal signal to target class images, causing the model to associate the signal with that class. Zhong et al. (2020) increased the stealthiness of attacks by adding imperceptible changes to a small portion of the training data, resulting in clean-label attacks where images appear to match their labels. Saha et al. (2020) extended this approach to transfer learning scenarios by injecting poisoned samples into the fine-tuning dataset. Ning et al. (2021) proposed disguising poisoned images as noisy ones, further complicating their detection in the training data. Inspired by gradient matching techniques from Geiping et al. (2020), Souri et al. (2022) introduced the Sleeper Agent

attack, which embeds triggers during neural network training from scratch in a black-box setting, where the attacker is unaware of the model architecture and training routine.

Despite extensive research dedicated to embedding a single trigger within models, the exploration of multi-trigger backdoor attacks in image classification remains limited, encompassing efforts that focus on methods to generate input-specific triggers (Nguyen & Tran (2020); Doan et al. (2022)) as well as those utilizing multiple static triggers (Xue et al. (2024); Gong et al. (2021)). This study introduces Gradient Storm, an enhanced version of the Sleeper Agent attack (Souri et al. (2022)), which is capable of injecting multiple backdoor triggers into neural networks simultaneously during the training phase. Our main contributions are:

- Stronger Noisy Gradients: We strengthen the Sleeper Agent attack introduced by Souri et al. (2022) through dividing the optimization process into multiple rounds each corresponding to a different region of the parameter space.
- Multiple Triggers, Targets, and Attack Types: Our attack poisons a victim model with backdoors of different kinds, each activated by its own trigger, and corresponding to a unique source and target pair.

This paper is organized as follows. Section 2 discusses related work on backdoor attacks and defenses. Section 3 describes our attack scheme in detail. Section 4 presents the experimental results demonstrating the effectiveness of our proposed method. Finally, Section 5 concludes our work.

## 2 RELATED WORK

Data poisoning attacks initially aimed at reducing the classification performance of traditional machine learning models, such as Support Vector Machines (SVM) (Biggio et al. (2012)). Unlike these indiscriminate attacks, targeted attacks focus on altering model outputs for specific test instances (Mozaffari-Kermani et al. (2014); Nelson et al. (2008)). Subsequent works have introduced clean-label poisons, which appear visually normal, to minimize the likelihood of detection (Shafahi et al. (2018)). In the context of targeted clean-label attacks, poisons have been specifically designed to manipulate models in both transfer learning (Zhu et al. (2019); Aghakhani et al. (2021)) and from-scratch training scenarios (Huang et al. (2020); Jagielski et al. (2021); Geiping et al. (2020)).

Backdoor attacks are a targeted data poisoning approach, designed to bind a static pattern (known as the *trigger*) to a target class in classification tasks. Researchers initially considered the trigger to be another image blended with (Chen et al. (2017)) or patched to (Gu et al. (2019); Wenger et al. (2021)) the original one. Subsequent studies explored the use of less conspicuous triggers to activate the backdoor, including weak sinusoidal signals (Barni et al. (2019)), reflections (Liu et al. (2020)), a warping effect (Nguyen & Tran (2021)), or a color space shift for all pixels (Jiang et al. (2023)). Research has also been directed towards increasing the variety of triggers. Gong et al. (2021), drawing on the mathematical formulation of the multi-trigger attack by Ji et al. (2018), have developed a method utilizing multiple triggers, each designed to activate specific neurons in outsourced cloud environments. Furthermore, researchers have proposed an alternative type of attack involving generative networks capable of producing triggers dynamically for any given input, thereby causing it to be classified into any desired class (Nguyen & Tran (2020); Doan et al. (2022)). These methods result in a lack of control over the trigger and potentially significant computational expense for designing the triggers. Recently, Xue et al. (2024) employed DCT Steganography to embed a text trigger in the channels of an RGB image. Researchers have also investigated multi-trigger backdoor attacks in the context of graph neural networks (Wang et al. (2024); Xu & Picek (2023)), neural code models (Li et al. (2023)), and deep image compression networks (Yu et al. (2023)).

To counter backdoor attacks, the most elementary defense mechanism involves identifying and excluding poisoned data from the training dataset, often requiring prior knowledge of the proportion of poisoned samples. For instance, Chen et al. (2019) proposed clustering samples to identify poisoned clusters, while Tran et al. (2018) used singular value decomposition to define outliers in the feature space. Researchers have also developed methods to remediate backdoored models, such as pruning neurons with abnormal activations and fine-tuning (Liu et al. (2018a)), reconstructing and mitigating triggers (Wang et al. (2019)), and using knowledge distillation to focus on original features (Li et al. (2021b)). Additionally, Wu & Wang (2021) utilized clean data to reveal trigger patterns and prune related neurons, and Zeng et al. (2022) formulated backdoor removal as a minimax problem solved

through unlearning. Robust training algorithms are another approach to mitigating backdoor attacks. Li et al. (2021a) increased the loss of misleading samples to prevent backdoor learning, while Hong et al. (2020) used differentially private SGD, and Borgnia et al. (2021) suggested data augmentations preserving differential privacy. Liu et al. (2022) added noise to training data to perturb examples without performance degradation, and Gao et al. (2023) developed an adaptive data-splitting method. These approaches are generally effective but often assume access to clean data or control over the entire training process, conditions that may not always be feasible.

## 3 METHOD

Following the threat model introduced by Gu et al. (2019), we consider an *attacker* who supplies a *victim* with poisoned data for the purpose of training a model from scratch. Given that the victim may have the ability to inspect the data for anomalies with the assistance of expert analysts, it is crucial for the attacker to ensure that the data does not appear to be conspicuously manipulated. Therefore, inspired by Souri et al. (2022), we assume that the attacker is restricted to making minor perturbations to a limited subset of the data. Although Souri et al. (2022) utilized these perturbations to embed a single trigger in the resulting model for activating the attack later, we extend their approach to embed multiple triggers. This extension allows for the potential activation of various types of attacks during inference. Building on the assumptions in Souri et al. (2022), we assume that the attacker does not have knowledge of the architecture of the final model or the learning algorithm employed. Consequently, we design the poisons using a surrogate model. In Section 4.3, we demonstrate that these poisons can effectively embed a backdoor in models with different architectures.

### 3.1 PROBLEM STATEMENT

Consider a surrogate model $F$ with parameters $\theta$ and a loss function $\mathcal{L}$. We address the optimization of perturbations $\Delta_1, \Delta_2, \ldots, \Delta_N$ to be added to the training dataset $D = \{(x_i, y_i)\}_{i=1}^N$. In this context, we are dealing with a multi-class classification problem, where each data point $(x_i, y_i)$ comprises a feature vector $x_i \in \mathbb{R}^d$ and a corresponding label $y_i \in \{1, 2, \ldots, C\}$. The objective is to solve the following optimization problem:

$$
\min_{\Delta \in L} \sum_i \mathbb{E}_{(x,y) \sim \mathcal{D}_i} \left[ \sum_j \sum_{k=1}^{n(i,j)} \mathcal{L}\left(F(\tau_{ij}^k(x); \theta(\Delta)), y_j\right) \right]
$$
$$
\text{subject to} \quad \theta(\delta) \in \arg\min_\theta \sum_{(x_l, y_l) \in D} \mathcal{L}\left(F(x_l + \Delta_l; \theta), y_l\right),
$$
(1)

where $\Delta_l$ denotes the $l$-th row of $\Delta$ and

$$
L = \left\{ \Delta \in \mathbb{R}^{N \times d} \mid \|\Delta_p\|_\infty \leq \epsilon \quad \forall p \in \{1, 2, \ldots, N\}, \ \delta_i = 0 \quad \forall i > B \right\}
$$

sets the limits on the magnitudes of perturbations, while indicating that only $B$ images may be perturbed. The attacker may consider $n(i,j)$ triggers each causing an image belonging to a source class $i$ to be identified as an image of a target class $j$, where $n(i,j) \geq 0$ for all values of $i, j \in \{1, 2, \ldots, C\}$. The function $\tau_{ij}^k$ applies the $k$-th trigger, causing that effect on an input image $x$ derived from the distribution of the source class $\mathcal{D}_i$. Note that for each pair of distinct triples $(i, j, k)$ and $(i', j', k')$, the functions $\tau_{ij}^k$ and $\tau_{i'j'}^{k'}$ may correspond to completely different types of triggers, such as a blended trigger and a patched one.

### 3.2 GRADIENT STORM

The problem posed in eq. 1 constitutes a bilevel optimization problem, as the perturbations $\Delta$ that minimize the upper-level objective are contingent upon the model parameters $\theta$, which are themselves determined by the minimization of the loss on the perturbed training data (the lower-level objective). To solve this challenging problem, following Souri et al. (2022), we employ the gradient matching technique, which has also been empirically validated as effective across multiple areas, including zero-shot learning (Sariyildiz & Cinbis (2019)), dataset condensation (Zhao et al. (2020)), domain generalization (Shi et al. (2021); Wang et al. (2023)), and condensing graphs (Jin et al. (2022)). It involves aligning the training gradients with a specified objective.

Prior to employing this methodology, we decompose the upper-level objective with respect to the source class $i$, target class $j$, and each trigger applier function $\tau_{ij}^k$ where $k \in \{1, 2, \ldots, n(i,j)\}$. This decomposition is based on the attacker's goal of causing images from the source class to be misclassified as belonging to the target class upon the application of the trigger applier function. Fixing the triple $(i, j, k)$ results in the following objective:

$$\mathcal{L}_{adv}(i,j,k) = \mathbb{E}_{(x,y)\sim\mathcal{D}_i} \left[ \mathcal{L} \left( F(\tau_{ij}^k(x); \theta), y_j \right) \right], \tag{2}$$

which is analogous to the adversarial objective examined by Souri et al. (2022), specifically for the case involving a single trigger to transition from one source class to a unique target class. Minimizing this objective requires $\theta$ to be known. Since we have not optimized the perturbations yet, we consider $\Delta$ to be a matrix of all zeros at the beginning and train $F$ on clean training data to obtain the initial $\theta$. Then we can estimate $\mathcal{L}_{adv}(i,j,k)$ over $\mathcal{P}$ training samples $\{(x_u, y_i)\}_{u=1}^{\mathcal{P}} \subset D$ belonging to the source class $i$:

$$\hat{\mathcal{L}}_{adv}(i,j,k) \overset{\text{def}}{=} \frac{1}{\mathcal{P}} \sum_{l=1}^{\mathcal{P}} \mathcal{L}(F(\tau_{ij}^k(x_u); \theta), y_j) \tag{3}$$

Similar to eq. 3, it is possible to estimate the gradient of the adversarial loss with respect to the model parameters $\theta$:

$$\hat{\nabla}_\theta \mathcal{L}_{adv}(i,j,k) \overset{\text{def}}{=} \frac{1}{\mathcal{P}} \sum_{l=1}^{\mathcal{P}} \nabla_\theta \left( \mathcal{L}(F(\tau_{ij}^k(x_u); \theta), y_j) \right) \tag{4}$$

If concerns about human inspection and the stealthiness of the attack were disregarded, we would directly inject $\{(\tau_{ij}^k(x_u), y_j)\}_{u=1}^{\mathcal{P}}$ into the training data. It could be argued that, assuming the user employs a gradient descent-based approach to train the model, the impact of this injection would be analogous to utilizing the gradient estimate in eq. 4 to update the model parameters $\theta$. However, since this approach is not feasible, we instead introduce minimal perturbations to a subset of training samples belonging to the target class $j$. Assuming we are permitted to perturb at most $B_{ij}^k$ such samples to implement the attack corresponding to the triple $(i, j, k)$, we distribute these perturbations across multiple sets of model parameters $\theta_1, \theta_2, \ldots, \theta_R$. This is achieved by iteratively designing $\lfloor \frac{B_{ij}^k}{R} \rfloor$ perturbations and retraining the model using the perturbed training data. Consequently, the perturbations are dispersed over a wider region of the parameter space compared to the scenario where all perturbations are designed according to a single set of parameters $\theta$.

Each step of this process involves taking $q \overset{\text{def}}{=} \lfloor \frac{B_{ij}^k}{R} \rfloor$ training samples $\{(x_v, y_j)\}_{v=1}^q$ belonging to the target class $j$ and adding perturbations to them such that their training gradient approximates the one given in eq. 4. This gradient could be again estimated as:

$$\hat{\nabla}_\theta \mathcal{L}_{train}(q,j) \overset{\text{def}}{=} \frac{1}{q} \sum_{v=1}^q \nabla_\theta \left( \mathcal{L}(F(x_v + \Delta_v; \theta), y_j) \right) \tag{5}$$

While fixing all other parameters, we design perturbations $\{\Delta_v\}_{v=1}^q$ to align $\hat{\nabla}_\theta \mathcal{L}_{adv}(i,j,k)$ with $\hat{\nabla}_\theta \mathcal{L}_{train}(q,j)$ through minimizing a similarity loss $\mathcal{A}$ defined as follows:

$$\mathcal{A}(\Delta, \theta, i, j, k) \overset{\text{def}}{=} 1 - \frac{\hat{\nabla}_\theta \mathcal{L}_{adv}(i,j,k) \cdot \hat{\nabla}_\theta \mathcal{L}_{train}(q,j)}{||\hat{\nabla}_\theta \mathcal{L}_{adv}(i,j,k)|| \cdot ||\hat{\nabla}_\theta \mathcal{L}_{train}(q,j)||} \tag{6}$$

We consider all pairs of source and target classes $(i, j)$ for which there is at least one trigger, i.e., $n(i,j) > 0$, and repeat the process described above for each triple in the set $\{(i,j,k)\}_{k=1}^{n(i,j)}$. We repeat the whole procedure of optimizing the matrix $\Delta \in \mathbb{R}^{N \times d}$ from the scratch $S$ times. Additionally, we use validation data consisting of unseen images that have been modified by trigger applier functions. At the end of each cycle, we calculate the percentage of these images that are classified according to the attackers' intentions, known as the attack success rate. We then select the $\Delta$ value corresponding to the highest cumulative attack success rate. Algorithm 1 and Figure 1 summarize our poison crafting approach.

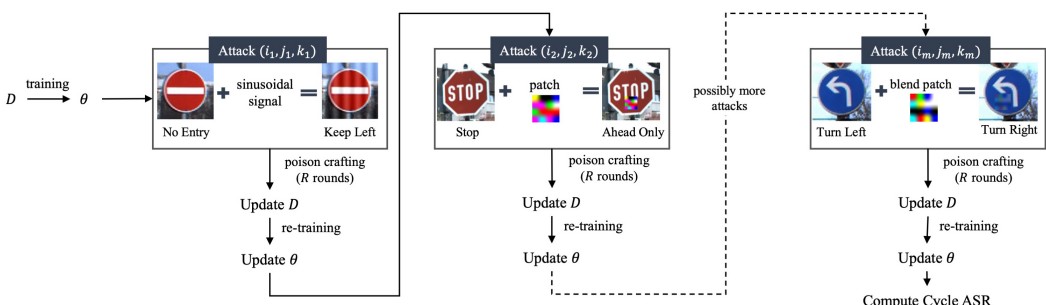

Figure 1: **Overview of a poisoning cycle in Gradient Storm:** The process initiates with the training of a model on a clean dataset $D$. Subsequently, multiple attacks are sequentially executed, each designed to introduce distinct triggers into the final model, such as a sinusoidal signal, a patched image, or a blended patch. These attacks specifically target and perturb the training images of particular classes (e.g., "Keep Left", "Ahead Only", "Turn Right") across $R$ rounds. After each attack, the perturbed dataset is employed to retrain the model, which is then subjected to the next attack in the sequence.

---

**Algorithm 1** Gradient Storm poison crafting procedure

---

**Require:** Training data $D = \{(x_i, y_i)\}_{i=1}^N$, Validation data with embedded triggers $V$, Number of triggers $n(i,j)$ for each pair of source and target indices $(i,j) \in \{1, 2, \ldots, C\}^2$, Trigger applier functions $\{\tau_{ij}^k\}_{k=1}^{n(i,j)}$ and poison budgets $\{B_{ij}^k\}_{k=1}^{n(i,j)}$ for each $(i,j)$-pair where $n(i,j) > 0$, Optimization cycles $S$, Cycle rounds $R$, Gradient Matching threshold $T$

1: Train a surrogate neural network $F(\cdot; \theta)$ on training data $D$
2: $D_o \leftarrow$ A copy of the original training dataset $D$
3: **for** $s = 1, 2, \ldots, S$ optimization cycles **do**
4:   $\text{ASR}_{\text{cycle}}(s) = 0$   (total ASR of the cycle)
5:   **for** $(i,j) \in \{1, 2, \ldots, C\}^2$ **do**
6:     **if** $n(i,j) > 0$ **then**
7:       **for** $k \in \{1, 2, \ldots, n(i,j)\}$ **do**
8:         **for** $r = 1, 2, \ldots, R$ cycle rounds **do**
9:           $q \leftarrow \lfloor \frac{B_{ij}^k}{R} \rfloor$
10:          $\mathcal{I} \leftarrow$ Indices of $q$ samples with label $y_j$ from $D$ with highest gradient norm that are not previously chosen in this cycle
11:          Randomly initialize perturbations $\Delta = \{\Delta_p\}_{p \in \mathcal{I}}$
12:          Compute the value of $\mathcal{A}(\Delta, \theta, i, j, k)$ and iteratively update $\Delta$ using the Adam optimizer until $\mathcal{A}(\Delta, \theta, i, j, k)$ is less than $T$
13:          $D \leftarrow \{(x_l, y_l)\}_{l \notin \mathcal{I}} \cup \{(x_h + \Delta_h, y_j)\}_{h \in \mathcal{I}}$
14:          Retrain $F$ on the poisoned training dataset $D$
15:          Update $\text{ASR}_{\text{cycle}}(s)$
16:        **end for**
17:      **end for**
18:    **end if**
19:  **end for**
20:  $D_s \leftarrow$ A copy of $D$
21: **end for**
22: Return $D_s$ for the optimization cycle $s$ having the best total ASR $\text{ASR}_{\text{cycle}}(s)$

---

## 4 EVALUATION

In this section, the effectiveness of our proposed attack is empirically evaluated using the CIFAR-10 (Krizhevsky et al. (2009)) and GTSRB (Stallkamp et al. (2011)) datasets as benchmarks. The Adversarial Robustness Toolbox (Nicolae et al. (2018)) serves as the foundation for our implementations. We use the patches introduced by Saha et al. (2020) as triggers, depicted in Figure 2. While training

all models, we use the same augmentations used by Souri et al. (2022). All experiments were conducted using a single NVIDIA TITAN RTX GPU to ensure consistent computational performance. For additional insights into the performance of different configurations, refer to the ablation study included in Appendix A.

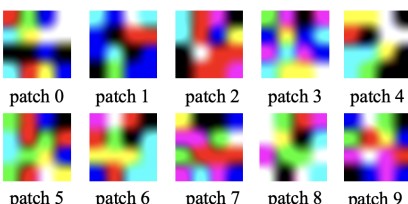

Figure 2: Patches used as triggers in our experiments, originally from Saha et al. (2020).

## 4.1 COMPARISON TO OTHER ATTACKS

We conduct a comparative analysis of the proposed Gradient Storm (GS) attack against several state-of-the-art backdoor attacks, including Blended (Chen et al. (2017)), Label-Consistent (LC) (Turner et al. (2019)), Refool (Liu et al. (2020)), Hidden Trigger Backdoor Attack (HTBA) (Saha et al. (2020)), and Sleeper Agent (SA) (Souri et al. (2022)). In all implementations, the $\ell_\infty$ norm of the perturbations is constrained to a maximum of 16/255. ResNet18 (He et al. (2016)) is utilized as the target model across all attack scenarios, with an $8 \times 8$ RGB image serving as the trigger. All models are trained for 80 epochs using the Stochastic Gradient Descent (SGD) optimizer with a momentum of 0.9, a weight decay of 5e-4, and an initial learning rate of 0.01. The batch size is set to 128, and all images are normalized to the range [0,1] before being standardized by subtracting the dataset mean and dividing by the standard deviation.

To ensure a fair comparison, ResNet18 is also employed as the surrogate model for both the SA and GS attacks. In the GS attack, we set $S = 4$ and $R = 2$, optimizing the poisons with $T = 0.006$, followed by model retraining. For the SA attack, we distribute the optimization process evenly across four retraining periods.

The HTBA attack, originally designed for transfer learning scenarios that require a fixed feature extractor, necessitates a distinct approach. The network is initially trained on clean data for 80 epochs. Poisons are subsequently generated based on the feature space representations extracted from the trained network. These poisons are then substituted for their benign counterparts in the original dataset, which is subsequently used to train a new model from scratch. The effectiveness of these poisons is evaluated in this from-scratch training scenario. To ensure fairness in the Refool attack, which requires a candidate reflection set, we utilize the same RGB image alongside three rotated versions at angles of 90, 180, and 270 degrees.

We present the experimental results using the following two metrics:

- Benign Accuracy (BA): The classification accuracy of the model on clean, unseen data during the inference stage.

- Attack Success Rate (ASR): The percentage of inputs containing the trigger, also unseen during training, that are successfully misclassified into the target class as intended by the attack.

The results of this comparative analysis are presented in Tables 1 and 2. For all experiments, the source and target classes were selected randomly.

## 4.2 RESISTANCE TO POISONING DEFENSES

In this section, we assess the effectiveness of various defense mechanisms, including general data poisoning defenses and those specifically designed to counter backdoor attacks, as applied to our method. The subsequent paragraphs provide a concise overview of each defense strategy.

| Attack | Blended | LC | Refool | HTBA | SA | GS (Ours) |
|---|---|---|---|---|---|---|
| BA (Poisoned Model) (%) | **90.17** | 90.01 | 90.29 | 89.7 | 90.02 | 90.06 |
| ASR (Poisoned Model) (%) | 91.5 | 2.3 | 2.93 | 81.46 | 89.73 | **99.76** |

Table 1: Comparing Gradient Storm with other Backdoor Attacks on CIFAR10. All attacks were performed with a poisoning budget of 500 samples.

| Attack | Blended | LC | Refool | HTBA | SA | GS (Ours) |
|---|---|---|---|---|---|---|
| BA (Poisoned Model) (%) | 93.44 | 92.23 | 94.35 | 92.19 | 93.38 | **94.36** |
| ASR (Poisoned Model) (%) | 38.67 | 1.41 | 20.36 | 69.29 | 58.19 | **84.25** |

Table 2: Comparing Gradient Storm with other Backdoor Attacks on GTSRB. All attacks were performed with a poisoning budget of 250 samples.

**Spectral Signatures**: Proposed by Tran et al. (2018), this method involves initially training a neural network on poisoned data to extract a learned representation for each input within each class. Singular Value Decomposition (SVD) is then performed on the representations of each class to compute an outlier score for each sample. The network is subsequently retrained after removing outliers, and this process is repeated over multiple iterations. The final model, which is expected to be free of backdoors, is then returned.

**Activation Clustering**: Chen et al. (2019) observed that while backdoor and target samples are classified identically by the poisoned network, the underlying reasons for this classification differ. Building on this observation, they clustered the learned representations of samples within each class and found that, for the target class of backdoor attacks, activations tend to separate into two clusters of approximately equal size. They then conducted cluster analysis to identify the poisoned cluster, which was subsequently removed from the training process.

**DeepKNN**: Peri et al. (2020) proposed a method that involves analyzing the nearest neighbors of each sample in the feature space and removing those for which the model assigns a label that differs from the majority of their neighbors. This approach has been empirically tested against targeted clean-label data poisoning attacks.

**Gradient Shaping**: Hong et al. (2020) observed that gradients computed in the presence of poisoned data exhibit significantly higher $\ell_2$ norms and orientations that diverge substantially from those of clean gradients. To address this, they employed Differentially Private SGD (DPSGD) to constrain gradient magnitudes and reduce discrepancies in their orientations during model training.

**Anti-Backdoor Learning (ABL)**: Li et al. (2021a) empirically identified a critical vulnerability in backdoor attacks: models tend to learn backdoored data significantly faster than clean data, with stronger attacks resulting in even more rapid convergence on the backdoored data. Leveraging this observation, they developed a method to isolate poisoned examples in the early stages of training and subsequently disrupt the correlation between backdoor examples and the target class in later training stages.

**DP-InstaHide**: Building on the work of Zhang et al. (2018), who introduced MixUp as a powerful data augmentation technique, Borgnia et al. (2021) demonstrated that combining these augmentations with random additive noise can effectively neutralize poisoning attacks with minimal accuracy trade-off. Additionally, they provided a theoretical proof that, under certain assumptions, their method satisfies differential privacy.

**Implicit Backdoor Adversarial Unlearning (I-BAU)**: Assuming access to a small set of clean data, Zeng et al. (2022) proposed an iterative algorithm to sanitize a poisoned model by formulating and solving a minimax problem between the defender and the attacker. Additionally, they conducted a theoretical analysis of the algorithm's convergence and demonstrated that the robustness achieved through this approach extends to unseen test data.

**Model Orthogonalization (MOTH)**: Tao et al. (2022) defined the $L^p$ norm of the smallest backdoor pattern that induces misclassification of images from a source class as those of a target class,

referring to this norm as the distance between the two classes. Their objective was to train an orthogonal classifier in which every pair of classes is maximally distant, a process they termed *model orthogonalization*.

We use the CIFAR-10 dataset and ResNet18 as the target model, employing the same hyperparameters as in Section 4.1, and apply defense strategies to the poisoned models from that section. Table 3 presents the results of applying these defense strategies to our proposed poison crafting method.

| Defense | ASR (%) | BA (%) |
|---|---|---|
| Spectral Signatures | 88.17 | 89.7 |
| Activation Clustering | 73.23 | 83.34 |
| DeepKNN | 97.97 | 87.62 |
| Gradient Shaping | 8.9 | 66.87 |
| ABL | **2.1** | 53.65 |
| DP-InstaHide | 6.47 | 67.26 |
| I-BAU | 39.93 | 90.67 |
| MOTH | 36.03 | **92.08** |

Table 3: The effectiveness of defense methods against the Gradient Storm attack

### 4.3 TRANSFERABILITY OF GRADIENT STORM

In this section, we examine the transferability of poisoned examples by assessing whether backdoors designed for a particular model can be successfully embedded in other models. Utilizing ResNet18 as a surrogate model and CIFAR10 as the training dataset, we evaluate the effectiveness of these poisons on several well-known architectures. All models are trained following the same procedure outlined in Section 4.1. The results are presented in Table 4.

| Model | ASR (%) | BA (%) |
|---|---|---|
| ResNet18 (Surrogate) | 99.60 | 90.15 |
| ResNet20 | 97.06 | 84.08 |
| ResNet34 | 99.8 | 90.53 |
| MobileNetV2 | 95.9 | 89.54 |
| VGG11 | 98.6 | 87.78 |
| VGG16 | 98.4 | 89.5 |

Table 4: Evaluation of the transferability of poisoned examples on CIFAR10 by testing backdoors crafted for ResNet18 across various well-known architectures.

### 4.4 MULTI-TARGET AND MULTI-TRIGGER SETTING

In this section, we empirically demonstrate that our method allows an adversary to execute multiple concurrent attacks, each characterized by a distinct source-target pair and trigger, while preserving the effectiveness of all attacks. In all experiments, ResNet18 is utilized as both the surrogate model and the target model, with CIFAR10 serving as the training dataset. In this series of experiments, we investigate different types of triggers. The simplest trigger type is a patch, which involves superimposing a small image onto a larger input image. The second type, known as a blended patch trigger, is formed by smoothly merging a small image with the larger input, blending the patch into the original content to reduce visual detectability. Inspired by Barni et al. (2019), we also consider sinusoidal signals as triggers, with horizontal signals defined by $v(i,j) = \Delta \sin(2\pi j f/m), 1 \leq j \leq m$ for row-wise oscillations, and vertical signals by $v(i,j) = \Delta \sin(2\pi i f/l), 1 \leq i \leq l$ for column-wise oscillations, at a given frequency $f$. In our experiments, we consider $\Delta = 0.1$ and $f = 5$. We present results for scenarios involving two and three simultaneous attacks. The outcomes for the two-attack scenario are detailed in Table 5, while the results for the three-attack scenario are shown in Table 6.

| Attack-1 | Attack-2 | BA (%) | ASR (%) | ASR-1 (%) | ASR-2 (%) |
|---|---|---|---|---|---|
| Bird → Deer (Horizontal Sinusoidal) | Automobile → Ship (Patch 0) | 89.19 | 96.2 | 95.2 | 97.2 |
| Airplane → Automobile (Blend - Patch 0) | Deer → Horse (Patch 1) | 90.2 | 92.95 | 92.6 | 93.3 |
| Airplane → Bird (Patch 5) | Truck → Airplane (Patch 8) | 89.25 | 99.3 | 99.7 | 98.9 |
| Dog → Frog (Blend - Patch 3) | Deer → Cat (Blend - Patch 9) | 90.16 | 91.7 | 96.6 | 86.8 |
| Cat → Ship (Vertical Sinusoidal) | Horse → Dog (Horizontal Sinusoidal) | 88.94 | 93.55 | 96.4 | 90.7 |

Table 5: Evaluation of the two-attack scenario using ResNet18 as the model and the CIFAR10 dataset. The first two columns list the source and target classes (source → target), with the specific trigger used to activate the backdoor shown in parentheses below each entry. The next two columns present the overall Benign Accuracy (BA) and Attack Success Rate (ASR), while the last two columns display the attack success rates for each individual attack.

| Attack-1 | Attack-2 | Attack-3 | BA (%) | ASR (%) | ASR-1 (%) | ASR-2 (%) | ASR-3 (%) |
|---|---|---|---|---|---|---|---|
| Deer → Dog (Patch 2) | Frog → Truck (Patch 5) | Deer → Dog (Patch 7) | 89.28 | 97.57 | 99.9 | 92.8 | 100.0 |
| Cat → Truck (Blend - Patch 4) | Frog → Bird (Blend - Patch 6) | Airplane → Truck (Blend - Patch 0) | 90.61 | 97.47 | 97.9 | 99.4 | 95.1 |
| Automobile → Horse (Patch 5) | Horse → Frog (Blend - Patch 2) | Truck → Dog (Horizontal Sinusoidal) | 89.12 | 91.93 | 100.0 | 93.7 | 82.1 |
| Dog → Cat (Patch 5) | Cat → Dog (Vertical Sinusoidal) | Truck → Ship (Horizontal Sinusoidal) | 89.41 | 90.13 | 99.9 | 78.8 | 91.7 |
| Deer → Bird (Patch 0) | Airplane → Frog (Patch 4) | Bird → Deer (Patch 9) | 90.15 | 93.2 | 98.4 | 89.1 | 92.1 |

Table 6: Evaluation of the three-attack scenario using ResNet18 as the model and the CIFAR10 dataset. The first three columns display the source and target classes (source → target), with the specific triggers used to activate the backdoors indicated in parentheses below each entry. The following two columns report the overall Benign Accuracy (BA) and Attack Success Rate (ASR), while the final three columns present the success rates for each individual attack.

## 5 CONCLUSION

In this study, we introduced Gradient Storm, a powerful and innovative method for executing multiple backdoor attacks in machine learning models with minimal data manipulation. Our approach demonstrates significant advancements in the design of adversarial poisons, allowing for precise control over specific regions of the parameter space and enabling the successful execution of multi-trigger attacks. Through comprehensive evaluation across two convolutional neural network architectures and two benchmark datasets, Gradient Storm has proven to be highly effective, achieving an attack success rate exceeding 90% in both single-trigger and multi-trigger scenarios. Additionally, the method shows strong resilience against a range of poisoning defense mechanisms, underscoring its potential impact on the security and trustworthiness of AI systems. These findings highlight the need for continued research into more robust defense strategies to counteract such sophisticated adversarial threats.

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

# A   APPENDIX

In this section, we conduct a series of experiments to evaluate the significance and contribution of each component of our algorithm. All experiments are carried out using the ResNet18 model as both the surrogate and target, with the CIFAR10 dataset serving as the training data. The results are averaged across various random selections of source and target classes.

## A.1   PATCH SIZE

We evaluate the influence of trigger size on attack success rate by experimenting with various sizes for both patch and blend triggers. All experiments are performed using the ResNet18 model as both the surrogate and target. In this section, poisons are generated for a single cycle and round. The results of these experiments are presented in Table 7.

| Trigger Type | Size | BA | ASR |
|---|---|---|---|
| Patch | $2\times 2$ | 90.96 | 0.3 |
| Blend | $2\times 2$ | 91.5 | 0.1 |
| Patch | $4\times 4$ | 90.09 | 11.1 |
| Blend | $4\times 4$ | 89.46 | 2.0 |
| Patch | $8\times 8$ | 88.52 | 73.2 |
| Blend | $8\times 8$ | 89.91 | 49.9 |
| Patch | $16\times 16$ | 90.53 | 95.8 |
| Blend | $16\times 16$ | 90.73 | 94.1 |

Table 7: Evaluation of the impact of trigger size on attack success rate. The table compares the success rates for various sizes of patch and blend triggers.

## A.2 POISON CRAFTING THRESHOLD

This section presents experimental results that illustrate how selecting an appropriate threshold influences the optimization objective in adversarial poison crafting. Figure 3 illustrates the threshold values used to terminate the poison crafting optimization process, along with the corresponding attack success rates achieved using the generated poisons.

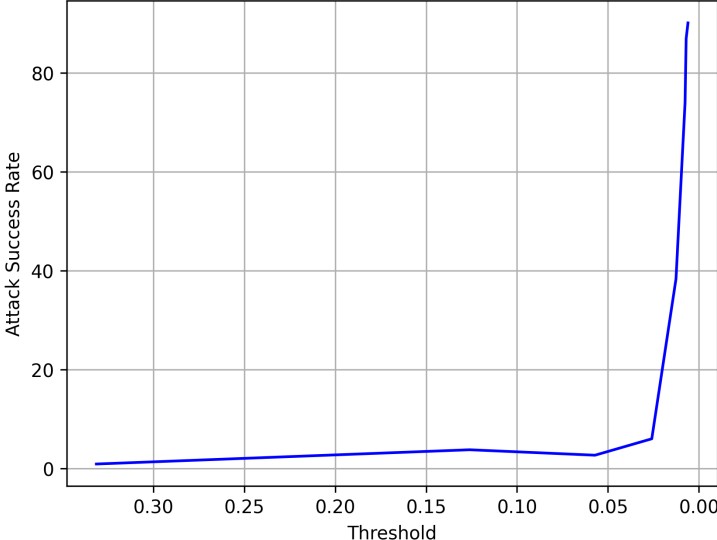

Figure 3: Threshold values and corresponding attack success rates for terminating the poison crafting optimization process.

