# OpenReview forum: "Gradient Storm: Stronger Backdoor Attacks Through Expanded Parameter Space Coverage"
_ICLR.cc/2025/Conference — Submitted to ICLR 2025_

### Official Review · Reviewer_SmHb · 2024-10-29

**Soundness:** 3
**Presentation:** 2
**Contribution:** 2
**Rating:** 3
**Confidence:** 3

**Summary:**

The paper improves existing attack methods and introduces "Gradient Storm," a more advanced data poisoning technique. The paper evaluate the proposed attack across various neural network architectures and two benchmark datasets, demonstrating its effectiveness.

**Strengths:**

The paper improves existing attack methods, and evaluate the proposed methods on different datasets & models.

**Weaknesses:**

Although the paper claims the proposed attack is more effective, but the results do not show that (Table 3). I also have other concerns as follows:

- Novelty: The changes seem minor compared to existing methods and the proposed method lacks convincing intuition. More explanation and comparative analysis of existing work (especially technical contributions) would be appreciated.

- Contribution: the paper mentions that one of the main contribution is “Multiple Triggers, Targets, and Attack Types”, which is not well justified. No clear demonstration of why paired target-trigger combinations matter. Also, simply implementing more attacks isn't a substantial contribution

- Evaluation settings: The paper only evaluates CIFAR and GTSRB datasets, and one trigger type. Evaluation on more datasets and triggers would be helpful.

- Evaluation results: the results in Table 3 shows that the proposed attack methods fails in at least 3 defense cases. It would be helpful to include the performance of other attacks against the defense methods (based on my experience, they fail less).

**Questions:**

N/A

---

> ### Author Response · Authors · 2024-11-28
>
> Thank you for your thoughtful review of our paper. Below, we offer responses to your questions and comments.
>
> > Evaluation results: the results in Table 3 shows that the proposed attack methods fails in at least 3 defense cases. It would be helpful to include the performance of other attacks against the defense methods (based on my experience, they fail less).
> >
>
> It is important to note that all defenses capable of neutralizing the attack also result in a decrease in benign accuracy. If the user is uncertain whether the model has been compromised, they may be reluctant to accept such a defense.

---

### Official Review · Reviewer_Q6eX · 2024-10-30

**Soundness:** 3
**Presentation:** 1
**Contribution:** 1
**Rating:** 3
**Confidence:** 5

**Summary:**

The article introduces the concept of gradient storms, a technique that simultaneously executes multiple backdoor attacks. The authors expand upon the existing optimization-based poisoning methods, enabling them to adapt to different attack targets and various triggers.

**Strengths:**

1. The article reexamines the existing poisoning framework and finds that it can be further expanded to a multi-trigger setting.
2. The method proposed in the article performs very well.
3. The method proposed in the article has good reproducibility.

**Weaknesses:**

1. The introduction chapter mainly consists of a literature review and lacks a clear presentation of its own contributions.
2. There is a problem of symbolic ambiguity with x_i and y_i in line 135. Does x_i refer to a sample or a feature? The subscript of y_i indicates both the index of the data and the index of the label.
3. There is a confusion between \delta and \Delta in lines 133 and 144.
4. The definition of D_i in line 150 is suggested to be elaborated.
5. Why does Formula 1 impose a norm constraint on the size of noise for any given case? This seems to be an unnecessary and uncommon term.
6. Tables 1 and 2 seem to compare different methods at all; they only conducted experiments using different triggers.
7. The contribution of the article is to expand SleeperAgent to a multi-trigger, multi-target setting, and there is still room for improvement in its contribution.

**Questions:**

What new methods does the article propose based on SleeperAgent?

---

> ### Author Response · Authors · 2024-11-28
>
> We are grateful for the time you spent reviewing our paper and have addressed your comments and questions as follows.
>
> > 2. There is a problem of symbolic ambiguity with x_i and y_i in line 135. Does x_i refer to a sample or a feature? The subscript of y_i indicates both the index of the data and the index of the label.
> >
>
> Thanks for mentioning this. x_i refers to a sample (not a feature), as it is stated that $x_i\in\mathbb{R}^d$ (line 135). We have replaced $y_i\in\{y_1,y_2,\dots,y_C\}$  with $y_i\in\{1,2,\dots,C\}$ (line 135) in the revised version.
>
> > 3. There is a confusion between \delta and \Delta in lines 133 and 144.
> >
>
> Thanks for pointing out. We have replaced $\delta = \{\delta_i\}_i^N$ with $\Delta_1,\Delta_2,\dots,\Delta_N$ in the revised version.
>
> > 5. Why does Formula 1 impose a norm constraint on the size of noise for any given case? This seems to be an unnecessary and uncommon term.
> >
>
> The constraint is to ensure that the poisoned samples remain visually similar to the original samples and are undetectable by human observers. Similar constraints were also used in previous work, including [1] and [2].
>
> > What new methods does the article propose based on SleeperAgent?
> >
>
> We have expanded the SleeperAgent attack to support multi-trigger attacks, where each attack can utilize a different type of trigger (e.g., Patch, Blend, and Sinusoidal).
>
> References:
>
> [1] Saha, A., Subramanya, A., & Pirsiavash, H. (2020, April). Hidden trigger backdoor attacks. In Proceedings of the AAAI conference on artificial intelligence (Vol. 34, No. 07, pp. 11957-11965).
>
> [2] Souri, H., Fowl, L., Chellappa, R., Goldblum, M., & Goldstein, T. (2022). Sleeper agent: Scalable hidden trigger backdoors for neural networks trained from scratch. Advances in Neural Information Processing Systems, 35, 19165-19178.

---

### Official Review · Reviewer_hiWg · 2024-11-01

**Soundness:** 2
**Presentation:** 2
**Contribution:** 1
**Rating:** 3
**Confidence:** 3

**Summary:**

The paper proposes Gradient Storm, a more effective version of the Sleeper Agent backdoor. The attack achieves its high effectiveness by designing perturbations that are distributed across the parameter space. This is accomplished by introducing *cycle rounds* where perturbations are optimized on disjoint subsets of the dataset. The method's effectiveness is validated against Sleeper Agent on CIFAR-10 and GTSRB.

**Strengths:**

- The writing is easy to follow.
- Sleeper Agent backdoors are an interesting and very hidden backdoor type that warrant more exploration.

**Weaknesses:**

- The novelty of the proposed extension to Sleeper Agent is limited.
- Limited datasets used in experiments. All experiments are done on CIFAR-10, except for one ASR experiment on GTSRB.
- The attack is very computationally intensive, requiring retraining a surrogate that approximates the attacked model (number of optimization cycles) * (number of cycle rounds) * (number of triggers) times. This implicitly assumes the attacker has access to very significant computational resources or is only able to attack small models.
- Standard ResNet-18 training uses random crop as a data augmentation [1], which is not use here. I suspect random crop would make this attack less effective.
- The paper does not describe what hyperparameters are used by each defense nor how those hyperparameters are derived. A maximally charitable evaluation of defenses would optimize hyperparameters against the attack and show how much clean data is required to remove the attack.
- The experiments in section 4.4 do not add very much beyond proving that a ResNet-18 has the capacity to learn 2-3 backdoors when training on CIFAR-10.
- It is unclear whether the increased ASR comes from the partitioning mechanism or from having multiple optimization cycles $S$ where the dataset containing the best randomized perturbations are returned.
- The paper does not provide an analysis on how different settings of cycle rounds $R$ and optimization cycles $S$ effects the success of the backdoor. The experiments section only examines one setting of these parameters without justifying how this setting is derived. This is a missed opportunity to demonstrate how valuable the proposed modification is to achieving a successful attack.

Minor
- Line 22 of algorithm 1 contains a typo.

**Questions:**

- Why do you use 8 retraining periods GA while only 4 for Sleeper Agent? What would be the results in tables 1 and 2 if 8 equally spaced retraining periods were used for the Sleeper Agent attack?
- What are the poisoning budgets used across the experiments ($B$ in algorithm 1)? How does using different poison budgets effect the ASR?

References

[1] He, Kaiming, et al. "Deep Residual Learning for Image Recognition," 2016.

---

> ### Author Response · Authors · 2024-11-28
>
> Thank you for the time and effort you’ve dedicated to reviewing our paper. We’ve included answers to your questions and comments here.
>
> > The attack is very computationally intensive, requiring retraining a surrogate that approximates the attacked model (number of optimization cycles) * (number of cycle rounds) * (number of triggers) times. This implicitly assumes the attacker has access to very significant computational resources or is only able to attack small models.
> >
>
> Please note that Small and medium-sized models are commonly used in real-world applications, such as edge devices, IoT, and embedded systems. For larger models, partial retraining or gradient approximation can be employed to mitigate the computational overhead.  The attack can also be applied in transfer learning scenarios. While the attack is designed for smaller models, as indicated in Table 3, the poisoned data can also be effective for other models, including larger ones (e.g., poisoned data designed for ResNet18 are also effective on ResNet34).
>
> > Standard ResNet-18 training uses random crop as a data augmentation [1], which is not use here. I suspect random crop would make this attack less effective.
> >
>
> We have used random augmentations (which include random crop) as in Sleeper Agent [1] and  Witches Brew [2] attacks. We have mentioned this in the revised version of the paper, lines 269 and 270.
>
> > Line 22 of algorithm 1 contains a typo.
> >
>
> Thanks for pointing out. The typo is fixed in the revised version.
>
> > Why do you use 8 retraining periods GA while only 4 for Sleeper Agent? What would be the results in tables 1 and 2 if 8 equally spaced retraining periods were used for the Sleeper Agent attack?
> >
>
> Our retraining periods differ from those used in the Sleeper Agent attack. In our approach, each retraining period is composed of two cycles, with half of the poisons designed using one set of model parameters and the other half using a different set. This contrasts with the retraining strategy in the Sleeper Agent attack, where all poisons are generated using a single set of model parameters.
>
> > What are the poisoning budgets used across the experiments (in algorithm 1)? How does using different poison budgets effect the ASR?
> >
>
> The poisoning budgets in Tables 1 and 2 are set at 500 and 250 samples, respectively. For the multi-trigger attacks, each attack from a source class to a target class was carried out with 500 samples. For example, to execute the attack described in the first row of Table 6, a total of 1500 samples were used (500 samples for each source-target attack).
>
> References:
>
> [1] Geiping, J., Fowl, L., Huang, W. R., Czaja, W., Taylor, G., Moeller, M., & Goldstein, T. (2020). Witches' brew: Industrial scale data poisoning via gradient matching. arXiv preprint arXiv:2009.02276.
>
> [2] Souri, H., Fowl, L., Chellappa, R., Goldblum, M., & Goldstein, T. (2022). Sleeper agent: Scalable hidden trigger backdoors for neural networks trained from scratch. Advances in Neural Information Processing Systems, 35, 19165-19178.

---

### Official Review · Reviewer_41Pn · 2024-11-10

**Soundness:** 2
**Presentation:** 3
**Contribution:** 2
**Rating:** 3
**Confidence:** 3

**Summary:**

This paper introduces a novel technique for enhancing backdoor attacks. Gradient Storm enables multiple backdoor triggers to be embedded within a model with minimal data manipulation. The attack leverages different regions of the model’s parameter space, facilitating multi-trigger, multi-target attacks. The paper demonstrates that Gradient Storm is effective across multiple architectures and datasets. Furthermore, the study shows that backdoors created in one model can transfer to other models.

**Strengths:**

Gradient Storm approach enables multi-trigger, multi-target backdoor attacks. The paper is well-written and easy to follow. The paper is supported by its experimental evaluations across various CNNs architectures and benchmark datasets,

**Weaknesses:**

The experimental section (4) lacks some comments on the results, i.e., the results are presented without analyzing if they support the claims presented.

Moreover, an evaluation of the trade-offs between the number of triggers and their stealthiness would strengthen the paper. The experimental design could benefit from a baseline comparison with simpler multi-trigger methods to better illustrate the advantages offered by Gradient Storm.

**Questions:**

- Can you please comment more on the implications of the results shown in Section 4?
- In light of Table 3, would you claim that Gradient Storm is able to bypass current defenses? Which hyperparemeters where chosen as for the defenses to do so? For instance, I-BAU and MOTH show a good performance against the attack
- Gradient Storm introduces various parameters, such as the number of triggers, poison budgets, and gradient matching thresholds. Could the authors provide guidance on tuning these parameters?
- Multi-trigger backdoors may introduce interaction effects between triggers. Can the authors provide an insight into how simultaneous or sequential activations of different triggers impact the model’s behavior?

---

> ### Author Response · Authors · 2024-11-28
>
> We appreciate the time you’ve taken to review our paper and have provided responses to your comments and questions below.
>
> > The experimental design could benefit from a baseline comparison with simpler multi-trigger methods to better illustrate the advantages offered by Gradient Storm.
> >
>
> To our knowledge, most prior methods for implementing multi-trigger backdoor attacks in image classifiers rely on optimization-based trigger design, resulting in unintelligible signals lacking human-interpretability (see Related Work, line 95). The only exception we found was a Steganography-based method mentioned in [1]. We suggest that comparisons with these methods may not be fully equitable, as our approach allows attackers to select meaningful, natural-looking triggers (e.g., a facial mole).
>
> > Can you please comment more on the implications of the results shown in Section 4?
> >
>
> Tables 1 and 2 demonstrate that Gradient Storm achieves more reliable backdoor embedding, as measured by Attack Success Rate (ASR) and Benign Accuracy (BA), compared to previous poisoning techniques, while allowing flexible trigger selection. Table 3 indicates that only defense methods significantly reducing BA can disrupt the attack, which makes the model's performance impractically low. Table 4 suggests that attackers do not need prior knowledge of the model architecture. Furthermore, Tables 5 and 6 show that multiple triggers for the same source or target classes can be embedded in the model without interfering with each other.
>
> > In light of Table 3, would you claim that Gradient Storm is able to bypass current defenses?
> >
>
> Table 3 suggests that disrupting the attack may lead to a significant drop in accuracy, rendering the model's performance ineffective. Consequently, users may be reluctant to accept the defense strategy, as they cannot be certain whether the training data was genuinely poisoned.
>
> > Gradient Storm introduces various parameters, such as the number of triggers, poison budgets, and gradient matching thresholds. Could the authors provide guidance on tuning these parameters?
> >
>
> The number of triggers and poison budgets are typically not chosen by the attacker, as they depend on the specific scenario. Most papers assume that the attacker can poison a small portion of the data (e.g., below 1% as in [2]).
>
> Section A.2 in the Appendix provides guidance on how selecting the appropriate threshold can influence the effectiveness of the resulting poisons.
>
> > Can the authors provide an insight into how simultaneous or sequential activations of different triggers impact the model’s behavior?
> >
>
> Thank you for the question. We have not specifically explored the interaction effects between simultaneously or sequentially activated triggers. This is an interesting area for future research, as such interactions could influence the model’s behavior, which we plan to investigate further.
>
> References:
>
> [1] Xue, M., Ni, S., Wu, Y., Zhang, Y., Wang, J., & Liu, W. (2022). Imperceptible and multi-channel backdoor attack against deep neural networks. arXiv preprint arXiv:2201.13164.
>
> [2] Souri, H., Fowl, L., Chellappa, R., Goldblum, M., & Goldstein, T. (2022). Sleeper agent: Scalable hidden trigger backdoors for neural networks trained from scratch. Advances in Neural Information Processing Systems, 35, 19165-19178.

---

### Meta-Review · Area_Chair_FMao · 2024-12-19

**Metareview:**

The paper proposes an extended version of Sleeper Agent backdoor attack called Gradient Storm. By desiging perturbations that distributed across the parameter space, the proposed attack can achieve multiple backdoor attack at the same time. The method's effectiveness is validated against Sleeper Agent on CIFAR-10 and GTSRB.

Strength:
1. The authors enabled the original SleepAgent settings to Gradient Storm, enabling the multi-trigger and multi-target backdoor attacks.
2. The paper's writing is clear.


Weakness:
1. Lack of novelty
2. The evaluation setting can be added.

Although this paper studies interesting backdoor settings proposed by SleeperAgent, the authors didn't clearly show their technical novelty in this paper. Therefore, the paper's novelty is limited and cannot be accepted by ICLR.

**Additional Comments On Reviewer Discussion:**

All the reviews show their strong concerns on the novelty of this paper and the authors haven't addressed all the concerns proposed by the reviewers during the rebuttal phase.

---

### Decision · Program_Chairs · 2025-01-22

Reject